Sucking lice in bandicoot rats with first complete description of Hoplopleura malabarica nymphs

Kozina Paulina 1 paulina.kozina@ug.edu.pl
Izdebska Joanna N. 1
http://orcid.org/0000-0003-3986-7659 Morand Serge 2 3 4
Ribas Alexis 5 6
1 Department of Invertebrate Zoology and Parasitology, Faculty of Biology, University of Gdansk , Gdańsk , Poland
2 IRL HealthDEEP, CNRS-Kasetsart University, Mahidol University , Bangkok , Thailand
3 Faculty of Veterinary Technology, Kasetsart University , Bangkok , Thailand
4 Department of Social and Environmental Medicine, Faculty of Tropical Medicine, Mahidol University , Bangkok , Thailand
5 Section of Parasitology, Department of Biology, Healthcare and the Environment, Faculty of Pharmacy and Food Sciences, Universitat de Barcelona , Barcelona , Spain
6 Institut de Recerca de la Biodiversitat (IRBio), Universitat de Barcelona , Barcelona , Spain
Mitchell Andrew
Electronic publication date: 2025 Oct 7
Publication date: 2025
Volume: 13
Electronic Location ID: e20115
Received 2025 Apr 22; Accepted 2025 Aug 29
Copyright: © 2025 Kozina et al.
Copyright year: 2025
Copyright holder: Kozina et al.
License: This is an open access article distributed under the terms of the Creative Commons Attribution License, which permits unrestricted use, distribution, reproduction and adaptation in any medium and for any purpose provided that it is properly attributed. For attribution, the original author(s), title, publication source (PeerJ) and either DOI or URL of the article must be cited.
License URL: https://creativecommons.org/licenses/by/4.0/

Keywords: Hoplopleura malabarica, Nymphs, Morphology characteristic, Checklist, Bandicota, Juvenile stages

Funding: The authors received no funding for this work.

==============================
Background

Studies of insect biodiversity and parasitism are often based on adult stages, as immature stages are poorly known and often cannot be identified to species level. However, sucking lice (Anoplura) are permanent, hematophagous parasites with single-host life cycles, making it possible to track the occurrence of all life stages. Only the complete identification of all life stages provides a full picture of parasitism, including infestation levels, parasite topography preferences on the host, and host specificity. The detection of different development stages on a host provides strong evidence that lice of a particular species are actively reproducing and completing their life cycle on that host, making full use of its resources. Conversely, the presence of adult lice alone, particularly when found sporadically, may suggest a failed or incidental attempt at host colonization rather than an established parasitic association.

Methodology

The description of the nymphal stages of Hoplopleura malabarica is based on specimens of sucking lice from the greater bandicoot rat Bandicota indica from Southeast Asia, specifically from the Vientiane area of Lao PDR. The study used morphometric analysis and scanning microscopy techniques.

Results

This study presents the first description of the nymphal stages of Hoplopleura malabarica, an oligoxenous parasite of rodents of the genus Bandicota. In addition, a global checklist of Anoplura parasitizing rodents of the genus Bandicota was provided.

Conclusions

The detection of different life stages of lice within the host confirms that they reproduce and develop on a given host species, fully utilizing its resources.

Introduction

Description of insect biodiversity, including parasitic insects, are typically based on imaginal stages. Many species have been described solely from adults, sometimes from only one sex. The immature stages of many species remain poorly documented, making accurate species-level identification challenging. In some cases, they inhabit different environments, requiring distinct methodologies for study. However, Anoplura are permanent parasites with single-host life cycles, making it possible to track the occurrence of all life stages within the host. All life stages of Anoplura feed on host blood, classifying them as hematophagous insects that can adversely affect host condition when infestation levels are high. Identification based only on imaginal stages is therefore incomplete and does not provide a full picture of parasitism, including aspects from the classical measures of parasitism to parasite ecology.

Many Anoplura species lack descriptions of their nymphal stages, making it extremely difficult or impossible to properly analyze infection levels, parasite topography within the host, and host specificity. The presence of different life stages within a host confirms that lice of a given species reproduce and develop on that host, fully utilizing its resources. In contrast, the mere detection of adult lice—especially occasional ones—may indicate only an attempt to colonize a new host rather than a successful parasitic relationship. To obtain a comprehensive understanding of lice parasitism and accurately analyze parasite–host relationships, it is essential to include information on host specificity. This also requires analyzing the composition of lice species within a host, which is only possible when immature stages are properly recognized and described. Identification of all stages of lice is necessary to determine the total level of infestation (the host’s parasite load), as well as to know the life cycle of lice.

An interesting subject for such studies is Hoplopleura malabarica (Kim, 1968), an oligoxenous parasite found exclusively on rodents of the genus Bandicota, including B. bengalensis (Gray, 1835), B. savilei (Thomas, 1916), and B. indica (Bechstein, 1800; Durden & Musser, 1994). Knowledge of and the ability to identify immature stages will enable future research on the full parasite–host relationships of Hoplopleura malabarica and its potential hosts. It will also make it possible to determine host preferences and distinguish between primary and additional hosts.

The previously undescribed nymphal stages of H. malabarica can be characterized in detail on the basis of morphological features visible on examination using light and scanning microscopy.

The aim of the current study was to make for a first time a detailed description of all nymphal stages of Hoplopleura malabarica. In addition, checklist of Anoplura parasitizing rodents from the genus Bandicota was prepared.

Materials and Methods

Material of sucking lice and host

The host specimens originated from Southeast Asia, specifically from Vientiane province of Lao PDR.

Bandicota indica is not a protected species and is not listed under Convention on International Trade in Endangered Species of Wild Fauna and Flora (CITES). The International Union for Conservation of Nature (IUCN) Red List classifies the greater bandicoot rat as Least Concern (LC). Rodents were handled in accordance with the guidelines of the American Society of Mammalogists and the European Union legislation (Directive 86/609/EEC). The trapping campaign was validated by national health authorities as part of a rodent-borne disease survey. Ethical approval for investing rodents for ectoparasitism was granted by the National Ethics Committee for Health Research (NECHR), Ministry of Health of Lao PDR, number 039/2016/NECHR.

Detailed methods for rodent collection and host identification are described elsewhere (Herbreteau et al., 2011). The study material comprised 12 specimens of the greater bandicoot rat (Bandicota indica, Rodentia) captured from the field, sacrificed according to the Animal Care and Use Committee of the American Society of Mammalogists (Sikes, 2016). The skin of the entire rodents was removed in the field and preserved in ethanol, posteriorly were keep in the Laboratory of Parasitology, Faculty of Pharmacy, University of Barcelona (Spain) until it was surveyed for the presence of sucking lice.

The search of sucking lice were performed under a transmission microscope by combing through the whole body hair, strand by strand. All detected individuals were collected with the aid of tweezers and stored in separate containers 70% ethyl alcohol. For the morphometrical analyses, individuals were placed in polyvinyl-lactophenol solution (Kadulski & Izdebska, 2006) and posteriorly examined under a light microscope (Olympus CX 40). All measurements in this work are given in millimeters (mm).

Sucking lice individuals intended for analysis with scanning electron microscopy were subjected to a series of alcohols (80–100%) and then dried (mix of ethyl alcohol and hexamethyldisilazane (HMDS)) in proportions: 1:3, 1:1, and 3:1. Finally, the specimens were transferred to pure HMDS and placed in an incubator for 24 h (37 °C) (Murtey & Ramasamy, 2016). Next specimens were stuck to double-sided copper tape (by Mierzejewski Materiały Samoprzylepne) and positioned on the table of Field Emission Scanning Electron Microscope JSM—7800F (manufacturer JEOL; stocked in Department of Materials Engineering and Bonding—Faculty of Mechanical Engineering, Gdansk University of Technology, Gdansk, Poland).

Many studies using the mentioned technique (in combination with traditional methods of analysis) show that it is highly effective in morphological analyses, especially in imaging key taxonomic features of sucking lice (Anoplura) (Kohn & Štěrba, 1987; Soler Cruz & Martín Mateo, 2009; Castro, Romero & Dreon, 2002; Leonardi et al., 2012; Kozina & Izdebska, 2021; Kozina, Izdebska & Kowalczyk, 2021; Kozina, Izdebska & Łopucki, 2022).

Hoplopleura malabarica was the only louse species observed on Bandicota indica, eliminating any possibility of misidentification of nymphs.

The louse specimens were deposited in the Scientific Collection of the Department of Invertebrate Zoology and Parasitology (UGDIZP), which is part of the Gdańsk Centre for Biological Resources of the University of Gdańsk, Gdańsk, Poland (Zhang, 2018).

Names and abbreviations of individual bristles and body parts follow Kim (1966) and Kim & Ludwig (1978):

ACHS anterior central head setae;

AcS accessory setae;

ADHS accessory dorsal head setae;

AHS apical head setae;

AnS anal setae;

CS clypeal setae;

DAHS dorsal anterior head setae;

DCAS dorsal central abdominal setae;

DMsS dorsal mesothoracic setae;

DMtS dorsal metathoracic seta;

DPTS dorsal principal thoracic setae;

DPtS dorsal prothoracic seta;

GP gular plate;

ISHS inner sutural head setae;

MAS major abdominal setae;

MHS marginal head setae:

AMHS anterior marginal head setae,

MMHS middle marginal head setae,

PMHS posterior marginal head setae;

OS oral setae;

OSHS outer sutural head setae;

PAS preantennal setae;

PCHS posterior central head setae;

PDHS posterior dorsal head setae;

VMHS ventral marginal head setae;

VPHS ventral principal head seta.

Global checklist of Anoplura parasiting rodents from the genus Bandicota

The world checklist of Hoplopleura species observed on rodents belonging to the genus Bandicota is based on publications from 1923 to 2014 (Ferris, 1923; Werneck, 1954; Johnson, 1959; Durden & Musser, 1994; Cardozo-de-Almeida, Linardi & Costa, 1999; Dong et al., 2014), no relevant literature after 2014 was found. The scientific names, common names, geographical range and systematics were based following the Mammal Diversity Database (2023).

Results

The lice material included 29 specimens of Hoplopleura malabarica, consisting of five males, 15 females, two first-instar nymphs, two second-instar nymphs, and five third-instar nymphs. These lice were collected from five of the 12 examined specimens of B. indica (Table 1). Prevalence was 41.7% (5 of 12 hosts infected), mean intensity was 5.8 individuals and the range of intensity was 2–18 individuals.

Table 1 Details of examined rodent materials and collection numbers of found Hoplopleura malabarica from Vientiane province in Lao PDR.

Bandicota indica UGDIZP collection numbers and sex	Date of collection	Location	Hoplopleura malabarica collection numbers	
District	GPS data	
MRMBi499/2015 ♂	12.05.2015	Phonehong	18°17′01.0″N		
102°30′45.6″E	
MRMBi507/2015 ♀	13.05.2015	Phonehong	18°18′30.7″N	4 Females (UGDIZPMBindAHHmal1f; UGDIZPMBindAHHmal2f; UGDIZPMBindAHHmal3f; UGDIZPMBindAHHmal4f)	
102°30′52.2″E	
MRMBi516/2015 ♂	13.05.2015	Thulakom	18°18′30.7″N	2 Male (UGDIZPMBindAHHmal1m)	
102°30′52.2″E	
MRMBi517/2015 ♀	13.05.2015	Thulakom	18°18′30.7″N		
102°30′52.2″E	
MRMBi518/2015 ♀	13.05.2015	Thulakom	18°18′30.7″N		
102°30′52.2″E	
MRMBi519/2015 ♂	13.05.2015	Thulakom	18°18′30.7″N		
102°30′52.2″E	
MRMBi522/2015 ♂	14.05.2015	Phonehong	18°21′21.0″N	1 Female (UGDIZPMBindAHHmal5f);
2 Nymphs third instar (UGDIZPMBindAHHmal2N3; UGDIZPMBindAHHmal3N3)	
102°27′21.2″E	
MRMBi527/2015 ♂	14.05.2015	Thulakom	18°18′30.7″N		
			102°30′52.2″E	
MRMBi528/2015 ♀	14.05.2015	Thulakom	18°18′30.7″N	2 Females (UGDIZPMBindAHHmal14f; UGDIZPMBindAHHmal15f)	
102°30′52.2″E	
MRMBi529/2015 ♂	15.05.2015	Phonehong	18°21′21.0″N		
102°27′21.2″E	
MRMBi553/2015 ♀	17.05.2015	Thulakom	18°17′31.2″N		
102°33′35.5″E	
MRMBi556/2015 ♂	18.05.2015	Thulakom	18°17′31.2″N	2 Nymphs first instar (UGDIZPMBindAHHmal1N1; UGDIZPMBindAHHmal2N1)	
102°33′35.5″E	2 Nymphs second instar (UGDIZPMBindAHHmal1N2; UGDIZPMBindAHHmal2N2)	
	3 Nymphs third instar (UGDIZPMBindAHHmal1N3; UGDIZPMBindAHHmal4N3; UGDIZPMBindAHHmal5N3)	
3 Males (UGDIZPMBindAHHmal2m; UGDIZPMBindAHHmal3m; UGDIZPMBindAHHmal4m)	
8 Females (UGDIZPMBindAHHmal6f; UGDIZPMBindAHHmal7f; UGDIZPMBindAHHmal8f; UGDIZPMBindAHHmal9f; UGDIZPMBindAHHmal10f; UGDIZPMBindAHHmal11f; UGDIZPMBindAHHmal12f; UGDIZPMBindAHHmal13f)	

Nymph body measurements are provided in Table 2 (in mm). Scanning microscopy revealed three variants of ornamentation in the form of scales shape: smoothly ended (U-shaped), sharp-ended (V-shaped), and needle-shaped setae (Kozina, Izdebska & Łopucki, 2022).

Table 2 Measurements of Hoplopleura malabarica nymphal body.

	Nymphs first instar	Nymphs second instar	Nymphs third instar	
		Average	Max	Min	Average	Max	Min	Average	Max	Min	
Length	Head	0.195	0.210	0.180	0.155	0.160	0.150	0.163	0.180	0.150	
Thorax	0.125	0.140	0.110	0.165	0.170	0.160	0.145	0.160	0.120	
Abdomen	0.555	0.600	0.510	0.485	0.550	0.420	0.710	0.740	0.690	
Total	0.875	0.950	0.800	0.805	0.860	0.750	1.018	1.060	0.980	
Width	Head	0.165	0.180	0.150	0.155	0.170	0.140	0.163	0.170	0.150	
Thorax	0.305	0.310	0.300	0.275	0.290	0.260	0.323	0.360	0.270	
Abdomen	0.500	0.520	0.480	0.420	0.480	0.360	0.663	0.760	0.560	

Description of nymphal stages

Nymph I (Figs. 1, 2)

Figure 1 Hoplopleura malabarica, dorsal and ventral view of nymph I.

Figure 2 Hoplopleura malabarica, ventral view of nymph I (SEM magnification ×120, scale bar 100 μm).

Head

Ventral side—No gular plate. The area of the future gular plate is covered by tubercles and U-shaped bristles. The area they occupy resembles a rhombus. Four AHS (two on each side). Two OS. Four VMHS (two on each side). Two additional bristles near the site of the future VPHS.

Dorsal side—No CS. Four DAHS, evenly spaced in a row. Two PAS on each side. AS and PCHS very faint. OSHS and ISHS present, spaced apart by the length of the bristles. ACHS is present, positioned close together. MMHS is closer to AMHS than PMHS. PDHS is thicker than DPTS, reaching the first thoracic segment. ADHS present.

Thorax

Ventral side—Evenly covered with V-shaped scales; no additional bristles.

Dorsal side—DPTS extends to the beginning of the abdomen. DPtS, DMsS, and DMtS present.

Abdomen

Shaped like an inverted water droplet. Visible undulation where pleural plates will develop. Segmentation is not visible. There are five pairs of DCAS visible along the central line of the body, those starting from the side of the thorax larger than those at the end. MAS, AnS, and AcS absent.

Nymph II (Figs. 3, 4)

Figure 3 Hoplopleura malabarica, dorsal and ventral view of nymph II.

Figure 4 Hoplopleura malabarica, dorsal view of nymph II (SEM magnification ×100, scale bar 100 μm).

Head

Ventral side—The area of the future gular plate is covered by tubercles and U-shaped bristles. The area they occupy resembles a rotated triangle. Four AHS (two on each side). Two OS. Two VMHS on each side. VPHS present. Dorsal side—Four DAHS (two on each side). PAS, PCHS, ACHS, and AS are visible. ISHS and OSHS are present. All MHS evenly spaced. PDHS extends to the end of the first thoracic segment. ADHS short, positioned above PDHS, extending to its beginning.

Thorax

Ventral side—Evenly covered with V-shaped scales.

Dorsal side—DPTS extends to the first abdominal segment. DMsS present next to the mesothoracic spiracle. DPtS and DMtS are present. The surface is mostly smooth, with sparse, wide U-shaped scales, particularly on the anterior part of the first segment.

Abdomen

Inverted water droplet shape, as in nymph I, but more tapered at the end (sometimes flat-topped). Segmentation is not visible. The entire abdomen is evenly covered with U-shaped scales, with rare V-shaped scales at the beginning and end of the abdomen. There are three pairs of DCAS visible along the central line of the body, those starting from the side of the thorax larger than those at the end. MAS, AnS, and AcS absent.

Nymph III (Figs. 5, 6)

Figure 5 Hoplopleura malabarica, dorsal and ventral view of nymph III.

Figure 6 Hoplopleura malabarica, ventral view of nymph III (SEM magnification ×120, scale bar 100 μm).

Head

Ventral side—The area of the future gular plate is covered by tubercles and U-shaped bristles. The area they occupy resembles inverted pentagon. Very convex scales present around the antennae, mouth opening, and middle of the head, though sparsely distributed. Four AHS. Four VMHS on each side. Two OS bristles. VPHS measuring 1/4 to 1/5 of head length. Numerous smaller (central and lower) and larger (lateral) nodular formations at the site of the future gular plate (GP), some with fine bristles, forming a rhomboid shape with a rounded lower part or an arrow-like pattern.

Dorsal side—Haustellum raised higher than the angles of the labrum. Four AHS (two on each side). Two CS. Four evenly spaced DAHS. Two PAS on each side. PCHS, ACHS, and AS present. ISHS and OSHS are spaced 1–1.5 bristle lengths apart. AMHS, MMHS, and PMHS evenly spaced apart. Postanntenal angles are sharp and angular. PDHS extends almost to the femur of the second pair of legs. ADHS present, positioned above PDHS.

Thorax

Ventral side—Evenly covered with V-shaped scales, with some U-shaped scales interspersed. No additional bristles.

Dorsal side—DPTS extends to the beginning of the abdomen. DPtS, DMsS (twice as long as the others), and DMtS present.

Abdomen

Barrel-shaped, rectangular from a dorsal view, tapering only at the very bottom. Segmentation is not visible. The entire abdomen is evenly covered with U-shaped scales, transitioning to V-shaped scales in the middle. There are four pairs of DCAS visible along the central line of the body, those starting from the side of the thorax larger than those at the end. MAS, AnS, and AcS absent.

Global checklist of Anoplura parasitizing rodents from the genus Bandicota

Three species of Anoplura were recorded on all known species of Bandicota (one species from the genus Hoplopleura and two from Polyplax). The distribution of sucking lice is restricted to Asian territories and is closely associated with their hosts. The exception is Polyplax spinulosa, a cosmopolitan parasite (Table 3).

Table 3 Anoplura species parasitizing rodents of the genus Bandicota with their geographic distributions.

Host	Host distribution	Anoplura species	Parasite distribution	Author	
Bandicota indica (Bechstein, 1800)
greater Bandicoot Rat	Bangladesh, Cambodia, China, India, Lao PDR, Myanmar, Nepal, Sri Lanka, Thailand, Vietnam and introduced in Malaysia	Hoplopleura malabarica	India, Sri Lanka, Thailand	Werneck (1954), Johnson (1959), Durden & Musser (1994), Cardozo-de-Almeida, Linardi & Costa (1999)	
Polyplax asiatica	China, Egypt, India, Iran, Myanmar, Pakistan, Taiwan, Tajikistan, Thailand	Ferris (1923), Durden & Musser (1994),
Dong et al. (2014)	
Bandicota bengalensis (Gray, 1835)
lesser Bandicoot Rat	Bangladesh, India, Myanmar, Pakistan, Sri Lanka and introduced in Indonesia and Malaysia	Hoplopleura malabarica	India, Sri Lanka, Thailand	Durden & Musser (1994)	
Polyplax spinulosa	cosmopolitan	Durden & Musser (1994)	
Polyplax asiatica	China, Egypt, India, Iran, Myanmar, Pakistan, Taiwan, Tajikistan, Thailand	Durden & Musser (1994)	
Bandicota savilei Thomas, 1916
Savile’s Bandicoot Rat	Restricted to Cambodia, Lao PDR, Myanmar, Thailand, Vietnam	Hoplopleura malabarica	India, Sri Lanka, Thailand	Johnson (1959), Durden & Musser (1994)	

Discussion

According our findings and the literature review, Hoplopleura malabarica appears to be a highly specific parasite, observed exclusively on rodents of the genus Bandicota. It is likely monoxenous to this genus, which is represented worldwide by only three rodent species (Mammal Diversity Database, 2023). The above data, reduce the risk of misidentification.

In contrast, two species of the genus Polyplax—P. asiatica and P. spinulosa—have been observed on the greater bandicoot rat. The morphology of immature stages of Hoplopleura and Polyplax differ significantly (among other things, the absence or presence of lateral lobes of paratergal plates; the shape of the abdomen; the presence of abdomen folding), facilitating accurate identification.

Considering all Hoplopleura species observed in neighboring countries of Lao PDR within the Indomalayan realm, the following species have been recorded (Johnson, 1959, 1964; Durden & Musser, 1994; Kazim et al., 2022): H. captiosa Johnson, 1960 on Mus caroli Bonhote, 1902;

H. diaphora Johnson, 1964 on Berylmys bowersi (Anderson, 1879);

H. dissicula Johnson, 1964 on Leopoldamys sabanus (Thomas, 1887), Maxomys whiteheadi (Thomas, 1894), Niviventer cremoriventer (Miller, 1900), Rattus argentiventer (Robinson and Kloss, 1916), R. baluensis (Thomas, 1894), R. rattus (Linnaeus, 1758), Sundamys muelleri (Jentink, 1879) and S. infraluteus (Thomas, 1888);

H. kitti Kim, 1968 on Berylmys bowersi;

H. malaysiana Ferris, 1921 on Leopoldamys sabanus and Sundamys muelleri;

H. pacifica Ewing, 1924 on Rattus argentiventer, R. exulans (Peale, 1848), R. norvegicus (Berkenhout, 1769), R. rattus and R. tiomanicus (Miller, 1900);

H. pectinata (Cummings, 1913) on Maxomys alticola (Thomas, 1888), M. rajah (Thomas, 1894), M. surifer (Miller, 1900), M. whiteheadi, Niviventer cremoriventer, N. niviventer (Hodgson, 1836), N. rapit (Bonhote, 1903);

H. rajah Johnson, 1972 on Maxomys surifer;

H. sicata Johnson, 1964 on Niviventer cremoriventer.

Given the number of species found in the Indomalaysian realm, the following are the features that differentiate H. malabarica nymphs from the others species.

The described first-instar nymphs of Hoplopleura malabarica can be morphologically differentiated from previous species reported in the Indomalayan realm. The presence of MAS in H. captiosa, H. diaphora, H. dissicula, H. kitti, H. pacifica, H. pectinata, and H. sicata. differs of H. malabarica that lacks of this structure. Third-instar nymphs of H. malabarica do not have tergal plaques, which are present in H. pacifica, H. pectinata, H. sicata, and H. rajah nymphs. The same applies to second-instar nymphs of H. dissicula and H. sicata. Additionally, MAS are present in all nymphal instars of H. diaphora but are absent in H. malabarica. Second-instar and third-instar nymphs of H. kitti have AnS and some dorso-central abdominal setae (DCAS), which are not found in H. malabarica. Second-instar H. pacifica also has AnS. No descriptions of H. malaysiana nymphs are available (Kim, 1966, 1968; Johnson, 1972).

In their work, Adhikary & Ghosh (1994) describe the nymphs of H. malabarica, but the taxonomical identity have to be taken with caution as is based on mass material, as many as 56 species of lice from a large area (India) are described. However, the description of H. malabarica and provided diagrams, indicate that the nymphs they examined do not belong to the above species. The description is quite laconic, while the diagrams lack elements important in identification (no scale, lack of most bristles, MAS bristles in questionable quantity). Below you can find a detailed reference to the characteristics of Adhikary & Ghosh (1994):

Nymph 1 “Approximately as long as wide”—it was not specified here whether this refers to the whole body or only the head, so it can’t be compared.

“Post antennal angles not developed”—it is impossible to confront this feature with current research.

“Antennal sensoria small and separate”—feature described in very general terms and difficult to confront.

“All typical head setae well developed and distinct”—current research characterizes head bristles, not only in terms of their presence, but also in terms of their length, thickness or position on the head. In addition, we observed the absence of clypeal setae (CS) on the dorsal side of the head.

“Ventral side of head with numerous scattered tubercles.”—current research confirms the presence of this feature.

“Thorax: MDTS one pair, small sized”–MDTP = DPTS in current studies. This bristle always appears as one pair. No definition of size or even its relation to other body parts.

“First pair of leg small with slender claw; second and third pair considerably larger with blunt claw, both legs are similar sized”—not very precise description.

“Abdomen: Devoid of segmentation”—current research confirms the presence of this feature.

“Ventral central setae 6 pairs; major abdominal setae one pair”—there are no ventral central setae and no major abdominal setae (MAS).

“Dorsal central setae absent”—currently, the presence of dorsal central abdominal setae (DCAS) has not been noticed either.

“Accessary setae 1 pair and anal setae 2 pairs”—in current studies there where no AcS and AnS observed.

Nymph 2 “Same as in nymph 1 except third pair of leg larger than second”—this feature is unclear. Nymphs, as they grow, increase in size. It is unclear whether the third pair of legs is larger relative to the other two or larger relative to those of a lower stage nymph.

“Abdomen with 2 pairs of major abdominal setae”—there are no MAS.

Nymph 3 “Same as in nymph 2 except major abdominal setae 3 pairs on each side”—there are no major abdominal setae (MAS).

Also authors provide information on number of the adults of H. malabarica was examined, but do not report the number of nymphs examined. The description of nymphs provided by Adhikary & Ghosh (1994) does not allow for correct species identification of the specimens they presented.

Conclusions

This study presents the descriptions of the nymphal stages of the Hoplopleura malabarica lice. The analysis of occurrence shows that it is a rare parasite, whose range of occurrence is limited to southern Asia (India, Sri Lanka, Thailand). H. malabarica is also a specific species of sucking lice that parasitizes only bandicoot rats (Bandicota indica-greater Bandicoot Rat, Bandicota bengalensis-lesser Bandicoot Rat, Bandicota savilei-Savile’s Bandicoot Rat). In addition to H. malabarica, two species of less specific lice of the genus Polyplax have been recorded on B. indica and B. bengalensis. Our study contributes to a better understanding of the genus Hoplopleura, providing morphological data that can be used to differentiate between the known species of this group of sucking lice.

Supplemental Information

Supplemental Information 1 Measurements of nymphs H.malabarica.

Additional Information and Declarations

Competing Interests

The authors declare that they have no competing interests.

Author Contributions

Paulina Kozina conceived and designed the experiments, performed the experiments, analyzed the data, prepared figures and/or tables, authored or reviewed drafts of the article, and approved the final draft.

Joanna N. Izdebska conceived and designed the experiments, performed the experiments, analyzed the data, authored or reviewed drafts of the article, and approved the final draft.

Serge Morand analyzed the data, authored or reviewed drafts of the article, and approved the final draft.

Alexis Ribas analyzed the data, authored or reviewed drafts of the article, and approved the final draft.

Animal Ethics

The following information was supplied relating to ethical approvals (i.e., approving body and any reference numbers):

The National Ethics Committee for Health Research (NECHR), Ministry of Health of Lao PDR approved the research (number 039/2016/NECHR).

Data Availability

The following information was supplied regarding data availability:

Raw data is available in the Supplemental File.

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
