# Peer review of "Sucking lice in bandicoot rats with first complete description of Hoplopleura malabarica nymphs"

_PeerJ, doi:10.7717/peerj.20115_

## Round 0.1 · original submission · Major Revisions

This manuscript has potential, however a number of serious issues must be addressed. The most important of these is the lack of consistent illustrations of all three nymphal stages. At a very minimum, the ventral surface of all three stages must be illustrated. Ideally the dorsal surfaces should be illustrated as well. The illustrations can be either light microscope or SEM photographs, or line drawings. Also, ideally both sexes should be illustrated where the sexes can be differentiated.

Also, the text must be changed to acknowledge that this manuscript is not the first to describe the nymphs of Hoplopleura malabarica, and the authors must discuss in detail why they regard the descriptions of Adhikary and Ghosh (1994) as insufficient.

I agree with the first reviewer that DNA sequence data would greatly strengthen this study, however I do not regard it as essential, provided a strong enough case is made for the veracity of the morphological evidence presented.

All reviewers provide useful suggestions and the authors must address all of them in their rebuttal letter.

Reviewer 1 ·

Basic reporting

I've reviewed Manuscript 117435v1 "Sucking lice in bandicoot rats with first description of Hoplopleura malabarica nymphs" by Kozina et al.

The authors' claim that this manuscript is the first description of Hoplopleura malabarica nymphs is not convincing by the evidence they presented. The three nymph stages Hoplopleura malabarica were already described in Adhikary and Ghosh (1994). The authors cited Adhikary and Ghosh (1994) in the current manuscript and questioned the validity of the nymph description presented in Adhikary and Ghosh (1994).

While I fully support the authors' effort in questioning the published work on this topic, I do not think the evidence the authors presented in the current manuscript is sufficient to reject the validity of Hoplopleura malabarica nymph description published in Adhikary and Ghosh (1994).

Experimental design

Using scanning microscopy method to produce high-resolution images is very helpful in studying the morphology of nymph stages of sucking lice. However, this method cannot replace plate illustration because scanning microscopy images may not be able to show all key, important identification features. The authors presented 3 scanning microscopy images (Figures 1-3) in the current manuscript. Figures 2 and 3 are clear but Figure 1 is not. The authors present the specimen in Figure 1 as nymph I, the specimen in Figure 2 as nymph II and the one in Figure 3 as nymph III. I have a slightly different view on these nymph stages. Based on the tail shape, I consider the specimen in Fig. 1 to be a later-instar male nymph, while Fig. 3 may represent a mid-instar female, as the posterior reproductive structures are already somewhat developed. Fig. 2, which shows the dorsal side, is mislabeled as ventral side in the manuscript. Fig. 2 is not directly comparable to the other two images, but the presence of a membranous covering on the posterior end suggests it could be a younger nymph, though its sex remains uncertain. I recommend the authors incorporate sex-based morphological differences where possible and clarify the staging with reference to posterior morphology. Also, both dorsal and ventral sides are needed when presenting the features of a nymph.

The study would be more robust and reliable if molecular sequencing data were provided to verify the identity of the different nymphal stages and adults. Integrating morphological and molecular evidence would enhance the completeness and credibility of the species identification and life stage association.

See Additional Comments section for more details.

Validity of the findings

As commented above, while I fully support the authors' effort in questioning the published work on the nymph description of Hoplopleura malabarica, I do not see sufficient convincing evidence in the current manuscript to reject the Hoplopleura malabarica nymph description published in Adhikary and Ghosh (1994). More thorough comparison between specimens and more convincing evidence are needed.

Additional comments

Line 1, 36, 72, 259, 270. Since the study by Adhikary and Ghosh (1994) has already described all three nymphal stages of Hoplopleura malabarica, the current manuscript cannot claim to be the first to do so. The present work may still hold value as an update or revision—particularly given the use of SEM imaging and potentially more detailed morphological comparisons—but the novelty claim should be removed from both the title and Introduction. If the authors believe that the identification in the 1994 study is questionable, this concern should be explicitly addressed, ideally by contacting the original authors for clarification or by re-examining the specimens used in that study if available.

Line 28. The phrase “imagines and nymphs” is not the most appropriate pairing of terms. For clarity and consistency in entomological terminology, it is recommended to use “adults and nymphs” or “imaginal and immature stages” instead. These pairings reflect standard usage and ensure terminological symmetry.

Line 104. The use of millimetres (mm) for reporting the body length of lice is acceptable. However, given the small size of Hoplopleura lice, particularly in their nymph stages, micrometres (µm) may be more appropriate when greater precision is required, especially for measurements below 1 mm.

For the “Description of nymphal stages”.
(1). It is recommended that the authors clearly specify the diagnostic rules or morphological criteria used to distinguish among the three nymphal instars of Hoplopleura malabarica. Clearly outlining these identification standards would enhance the transparency, accuracy, and reproducibility of the staging process.
(2). Although the study provides SEM images to support morphological descriptions, it is recommended that the authors include detailed line drawings with labelled diagnostic characters for all three nymphal stages. Such illustrations are a standard and valuable component of traditional morphological taxonomy, facilitating clearer comparisons and accurate species identification. Additionally, the current SEM images depict only the ventral sides of “Nymph I” and “Nymph III”, and the dorsal side of “Nymph II”. Since the morphological descriptions address both dorsal and ventral features, it is essential to provide SEM images of both views for each instar. Furthermore, a line drawing of the entire louse—illustrating one half as the dorsal side and the other as the ventral side with diagnostic characters clearly labelled—would significantly enhance the clarity and completeness of the morphological documentation.

Line 165. A numbering inconsistency in the tables. The supplementary table is labelled as "Table 1".

Line 178. In Fig. 1 (ventral view), segmentation is faintly visible, along with paratergal plates and associated spiracles. These features are not observable in Fig. 3, which may indicate that the specimen in Fig. 3 represents an earlier developmental stage. Since Fig. 2 shows the dorsal side, it is not directly comparable in this regard. This raises some uncertainty about the accuracy of instar classification in the figures. I suggest the authors clarify the criteria used for staging and ensure consistency across the illustrations.
In my view, the degree of development of the posterior reproductive structures may also serve as a useful supplementary criterion for distinguishing nymphal stages. Additionally, it is important to consider the sex of the nymphs when analysing morphological differences, as male individuals tend to be smaller than females and may exhibit distinct genital morphology. Therefore, when the sex is different, size should not be used as the primary criterion for staging the nymphs. Based on the tail shape, I consider the specimen in Fig. 1 to be a later-instar male nymph, while Fig. 3 may represent a mid-instar female, as the posterior reproductive structures are already somewhat developed. Fig. 2, showing the dorsal side, is not directly comparable to the other images, but the presence of a membranous covering on the posterior end suggests it could be a younger nymph, though its sex remains uncertain. I recommend the authors incorporate sex-based morphological differences where possible and clarify the staging with reference to posterior morphology.

Line 182, Figs 2. The image labelled as “Hoplopleura malabarica, ventral view of nymph II (SEM: magnification ×100)” appears to actually show the dorsal side. Please correct the labelling to avoid confusion.

Line 222-227. The two consecutive paragraphs on the host specificity and species identification of Hoplopleura malabarica partially overlap in content. Consider merging and condensing them to avoid redundancy and improve clarity.

Line 270-271. The sentence “Using scanning microscopy methods, the world’s first descriptions of the nymphal stages of the Hoplopleura malabarica lice have been created” is a bit awkward. Pls revise it.

Table 1. In the table title, “Hoplopleura species” should be reconsidered, as the table includes both Hoplopleura and Polyplax species. It would be more accurate to replace it with “Anoplura species” to reflect the broader taxonomic coverage.

In Table 2. The caption reads “Table 1 Details of examined rodent materials and collection numbers of found Hoplopleura malabarica from Vientiane province.” However, based on its position in the manuscript, this should be labelled as Table 2. Please correct the table numbering to ensure consistency and avoid confusion.

In Table 2, The date format “13. 05. 15” is inconsistent with the rest of the data, which uses the full four-digit year (e.g., “13. 05. 2015”). For clarity and consistency, please standardise all dates to the same format throughout the table.

In Table 3, The use of millimetres (mm) for reporting louse body length is acceptable; however, micrometres (µm) may be more appropriate if greater precision below 1 mm is needed.

In Table 3, The caption reads “Table 1 Measurement of Hoplopleura malabarica nymphal body.” However, based on its position in the manuscript, this should be labelled as Table 3. Please correct the table numbering to ensure consistency and avoid confusion.

·

Basic reporting

All fine in my opinion.

Experimental design

All fine.

Validity of the findings

All fine.

Additional comments

In this paper, the authors describe the 3 nymphal instars of the Asian sucking louse Hoplopleura malabarica Werneck, 1954 and provide lists of the lice recorded to parasitize Bandicota spp. rats along with their known geographical distributions. The authors are absolutely correct in stating that being able to identify immature stages of parasites permits a more comprehensive understanding of parasitism and allows for better understanding of host-parasite relationships including host specificity. Comparison of the immature stages of H. malabarica with those of other congeners from the region (lines 234-264) is especially useful.

The use of English is good throughout the paper with a few small exceptions, most of which I have marked (in pencil) in an accompanying file (only pages I marked are included).

General: Because sucking lice are ectoparasites, they occur “on” the host not “in” or “within” the host as I have marked. They also cause an “infestation” not an “infection” (correct on line 30, needs to be corrected on line 54).

Line1: Technically, this is not the first description of the nymphal stages of H. malabarica. Adhikary & Ghosh (1994) provided stylistic line-drawings of the nymphal stages. These differ in some aspects from the descriptions in this paper as noted below.

Line 116: I am unsure of the meaning of “successively deposited.” Can the authors please clarify?

Lines 169, 171 and others in the nymphal descriptions: “side” usually infers lateral not ventral or dorsal. I would therefore delete “side” in all ventral and dorsal entries (or replace it with “surface”).

Line 180: First instar nymphs of Hoplopleura usually have 1 pair of long posterior abdominal setae on each side. I do not see any in Figure 1 and I wonder if they could have broken off during specimen preparation for SEM. Could the authors please check? Adhikary & Ghosh (1994, p. 85) show a pair of long posterior abdominal setae on nymph I of H. malabarica but, as the authors of this paper state, those figures are stylistic. Interestingly, Adhikary & Ghosh (1994, p. 85) also show 2 pairs of long posterior abdominal setae on each side for nymph II and 3 pairs on each side for nymph III. The authors should comment on these differences compared to their nymphal descriptions for the same species.

Lines 169-219: Articles (“the” and “a”) are removed from classical taxonomic descriptions. They are retained here. Does the journal have a policy on this? I don’t mind either way but thought I should mention this.

Figure 2. The legend for this figure states that it is a ventral view but it is actually dorsal. Can the authors replace this SEM with a ventral SEM to conform with the other figures and so that more of the characters discussed in the description are shown? Actually, if specimens are available, showing both dorsal and ventral SEMs for all 3 nymphal instars would be useful (dorsal and ventral characters are covered in the descriptions). If not, I think just the ventral SEMs will suffice.

Reviewer 3 ·

Basic reporting

Basic reporting generally acceptable. However, I think the manuscript could be improved with additional tables and more clarity regarding the need for descriptions of nymphal life stages (including possible errors within the literature). See more comments below in “Additional Comments”

Experimental design

Experimental design can be improved. Missing from the manuscript is an acceptable description of field work for capturing host specimens. I think the authors can better explain the need of their research. The authors stress their use of SEM in their research, but they do not discuss the benefits/utility of SEM for describing nymphs. It would be helpful for the authors to expand on their use of SEM. See more comments below in “Additional Comments”

Validity of the findings

Validity of findings can be improved. I think in several places, the authors make statements that their data do not support. See additional comments above and well as in “Additional Comments”

Additional comments

General Comments
That abstract background states that all life stages of a parasite must be identified for a complete picture of parasitism. This is true for parasites with complex life cycles, using different hosts for different life stages. However, sucking lice have a direct life cycle with all stages occurring on one host. Thus, in this case, I don’t think nymphal stages need to be identified and described to better understand hos specificity, etc. (what the authors list at the end of the abstract background). Perhaps it would be best for the authors to better explain why describing nymphal forms is useful and needed? In this case, to more easily identify these life stages in the future which can help with understanding host associations, reduce misidentification, and perhaps better understand the biology of a particular louse species (how long at each life stage, survival, etc.). The authors make a good point in the introduction that I think could be valid for the abstract.

The materials and methods are unclear regarding how Bandicota specimens were obtained. Were specimens already in a museum/lab? Or did the authors collect the specimens? It seems like the latter, but how the authors have described these methods makes this point unclear. If the authors captured the animals in the field, information is missing as to how the animals were captured is missing. What kinds of traps, bait, etc.? It is also unclear when and how thoroughly the authors looked for lice on each host specimen. More information is needed. Lastly (and importantly), all specimens of both Bandicota and lice need to be installed in natural history collections for proper curation and care.

Methods, Results, and Discussion should all follow the name organization scheme. Personally, I think the order should be the nymphal stages of H. malabarica and then the host list. That is my personal preference, and my comments below reflect that preference. Whichever the authors choose is fine so long as the presentation order is the same throughout the manuscript (which I believe they have done).

Discussion: I think the discussion could be improved via the inclusion of tables for both the Hoplopleura associations and the morphological differences between nymphs of different Hoplopleura species. Some information presented in the discussion may be better placed in the introduction (Adhikary and Ghosh (1994) for example).

Results/Discussion: It would be nice if the authors can refer to the benefit of SEM used in their study. Why this method? What does it provide that a dissecting or compound microscope does not?

Specific comments and recommendations
Abstract: Background Line 25: I wonder if “imaginal stages” is the best term to use since lice do not undergo complete metamorphosis

Abstract: Background Line 27: Put “Anoplura” in parentheses: “..sucking lice (Anoplura) are…”

Abstract: Background Line 27: “stationary” implies the lice do not move. Consider using the term “permanent” or “obligate”

Abstract: Background Sentence beginning line 26: suggest editing to the second half of this sentence as it is awkward as written. Consider two sentences are describing sucking lice as hematophagous in the previous sentence. Also suggest changing words such as “within” to “on”

Abstract: Background Line 30: What do the authors mean by “parasite topography”. Again, choose a different word than “within” because lice are not found inside the body of the host. Lice occur on the exterior of their hosts

Abstract: Methodology: Consider two sentences where the second sentence states how the lice were observed/assessed.

Abstract: Results Line 38: consider changing “was compiled” to “is provided”

Abstract: Conclusions Line 39: consider changing the word “within”

Introduction, Line 43: Maybe replace “Analyses” with “Description” or other similar work. See comment above about use of “imaginal”

Introduction, Line 45: consider revision “Immature stages are often poorly known and cannot be identified accurately to species”

Introduction, Line 47: consider adding “(sucking lice)” after “Anoplura”. Authors should also probably describe Anoplura as insects and give the taxonomy (Insecta: Pscodea)

Introduction, Line 47: see comment above about use of “stationary”

Introduction, Line 48: see comments above about use of “within”

Introduction, Line 48: consider revision, something like: “Anoplura are hematophagous insects, where all life stages feeding on host blood and potentially impacting host health depending on infestation levels”

Introduction, Line 50: I don’t know that this sentence is needed, especially the second part of the sentence. Consider removing

Introduction, Line 54: see comment above; what is “parasite topography” and careful about use of “within”

Introduction, Line 55: see comments above about use of “within”

Introduction, Lines 55-58. This is a good point. The authors should incorporate this point into the abstract

Introduction, sentence beginning Line 58: consider “association” instead of “system”. But it is unclear what the authors mean by “range of host specificity”. The authors should consider breaking this sentence up into multiple sentences; it is unclear as written

Introduction, Line 62: Authors could consider defining “oligoxenous” for the readers

Introduction, Line 62: Authors could give taxonomy of the parasite. What family? Is this a well known family, etc.?

Introduction, Line 67: What do the authors mean by “auxiliary hosts”?

Introduction, Line 65: Consider revision to this sentence, perhaps two sentences? Or removing unnecessary information.

Introduction, when introducing the parasite: what are the known host associations for this particular species? If the authors place this information here, it will not be necessary to include in the next paragraph

Introduction, Line 70: consider removing this sentence.

Introduction, Line 72: If authors provide specific host species in previous paragraph, they do not need to state host species in this sentence.

Introduction, Line 74: This sentence could be removed. Authors could add some of this information (infestation and defining infestation) to the first or second paragraph

Introduction, Line 75: change tense to present. “In addition, we provide data on the checklist of Anoplura parasitizing….”. I don’t know that the word is “global” since Bandicota are not global in their distribution

Materials and Methods, Line 81: “on rodents” (not in). Consider rephrasing to “…observed on rodents belonging to the genus Bandicota”. Consider two sentences where the second sentence could read “To our knowledge, no studies of Bandicota lice have been published after 2014.”

Materials and Methods, Line 85: There is a new version of MMD that the authors should cite (citation year 2025)

Materials and Methods, Line 88: Please provide the catalog numbers of these specimens here or in supplementary material.

Materials and Methods, section beginning Line 87: this section is unclear. Did authors collect specimens and then deposit them in the collection? Or did authors solely examine specimens already in collections? As written, this section starts out implying the latter. But the section beginning line 92 implies the former. The authors should start out with information collecting in the field (proper permits, locality etc.), preservation, etc., and end with all specimens were cataloged. Also unclear: Is the Laboratory of Parasitology and natural history museum? If not, it is imperative that both host and louse specimens be deposited in natural history collections. And another unclear point: how were the animals captured? What kind(s) of traps were used? When were the traps set? How many traps? What kinds of bait? How were the animals euthanized? Please cite American Society of Mammalogists relevant documents (Sikes et al.)

Materials and Methods, Line 92-94: I’m not sure if the conservation status of the host species is needed here.

Materials and Methods, Line 100: Was each specimen preserved in a separate container? If not, how did the authors prevent cross-contamination of lice among host individuals? Can the authors provide more information about their search for lice on the host skins? It seems like fluid preservation would make it difficult to find lice. Or did the authors first look for lice on each host specimen, and then place the host specimen in fluid preservation after? Please clarify and provide specific details of how authors examined hosts (all over the hosts body? Only a portion of the host body, etc.).

Materials and Methods, line 113: How did the authors determine that H. malabarica was the only louse species found? Did they examine the adults? What identification (if any) did the authors use to verify their species identification.

Materials and Methods, Line 115: Sentence unclear. Please clarify.

Materials and Methods, Line 148: Would this section be better for the results?

Results, Lines 155-159: Present the checklist at the end. The authors main focus is H. malabarica, so they should give findings regarding this species first. Checklist last. Thus, present Table 2 before Table 1

Results: Authors should start with numbers. How many hosts had lice (5/12). How many lice per host (10-18) where host individual L0556 was the most useful, correct? Only this host had nymphs and adults.

Results, louse descriptions. Is it accurate to describe setae as “faint”? Would “thin” be a better term?

Results: for those hosts that only had nymphs, were those nymphs identical to the nymphs on host L0556? In other words, do all of the nymphs belong to the same species and that species is H. malabarica?

Discussion. The authors should not start out the discussion with host associations/host list. Instead, focus on the main focus of the manuscript, the nymphal stages of H. malabarica.

Discussion, Line 222: I don’t think the findings of authors did supports host specificity. The authors did not examine other potential hosts. Perhaps rephrase to focus on the results of their literature review and possible host specificity, at least to the genus Bandicota. The authors could also state that is seems that B. indica is only parasitized by one species of Hoplopleura, but additional research would be necessary examining hosts/literature/museum records from across the geographic range.

Discussion, Line 228: change “in” to “on”

Discussion, Lines 228-230: Can the authors elaborate how the morphology of Hoplopleura and Polyplax differ significantly? I think this would be important information for the readers.

Discussion, Lines 231-248: I think the authors should consider a table here, including geographic region and host taxonomy. Are all hosts Muridae like the Bandicota? Host taxonomy would be important to add.

Discussion, Line 249: Consider adding a new subheading here for readability

Discussion, Line 251: Remove period after “sicata”. I think the authors could phrase the morphological difference in other ways. Actually, another table here would be really helpful showing the morphological differences between species.

Discussion, Line 259: Please revise this section for clarity and typos. I also wonder if this section (or parts of this section) might be more suitable for the introduction. Possible error made by Adhikary and Ghosh (1994) of assigning non H. malabarica nymphs to H. malabarica. However, I think the authors are going to need to add more detail here. To what genus do the nymphs in Adhikary and Ghosh (1994) belong? Hoplopleura? Can the authors determine the correct species in Adhikary and Ghosh (1994)?

Discussion, Line 265: It would be nice if the authors could come back to some points they made in the introduction. This sentence should clarify differentiation of nymphs of known Hoplopleura species from this specific geographic area. Also, would this sentence be better for the “Conclusions” section.

Conclusions: It would be nice if the authors can refer to the benefit of SEM used in their study earlier in the manuscript. Why this method? What does it provide that a dissecting or compound microscope does not?

Conclusions, line 271: I disagree. The findings of the authors do not support that the parasite is rare. I suggest the authors remove this sentence.

Conclusions, line 273: Consider simply saying genus Bandicota instead of listing all species? Refer readers to the relevant table

Figure 2 looks like a dorsal view to me. Can the authors verify this is a ventral view?

Table 2 legend: consider revision for an informative legend, removing unnecessary words. I don’t think “of found” is needed.

---

## Round 0.2 · accepted · Accept

The reviewers' concerns have been adequately addressed.